# Isolation and Culture of Skin-Derived Differentiated and Stem-Like Cells Obtained from the Arabian Camel (*Camelus dromedarius*)

**DOI:** 10.3390/ani9060378

**Published:** 2019-06-20

**Authors:** Islam M. Saadeldin, Ayman Abdel-Aziz Swelum, Ahmed E. Noreldin, Hammed A. Tukur, Aaser M. Abdelazim, Mosleh M. Abomughaid, Abdullah N. Alowaimer

**Affiliations:** 1Department of Animal Production, College of Food and Agricultural Sciences, King Saud University, Riyadh 11451, Saudi Arabia; aswelum@ksu.edu.sa (A.A.-A.S.); tukurhammeda@gmail.com (H.A.T.); aowaimer@ksu.edu.sa (A.N.A.); 2Department of Physiology, Faculty of Veterinary Medicine, Zagazig University, Zagazig 44519, Egypt; 3Department of Theriogenology, Faculty of Veterinary Medicine, Zagazig University, Zagazig 44519, Egypt; 4Histology and Cytology Department, Faculty of Veterinary Medicine, Damanhour University, Damanhour 22511, Egypt; Ahmed.elsayed@damanhour.edu.eg; 5Department of Basic Medical Sciences, College of Applied Medical Sciences, University of Bisha, Bisha 61922, Saudi Arabia; aaserabdelazim@yahoo.com; 6Department of Biochemistry, Faculty of Veterinary Medicine, Zagazig University, Zagazig 44519, Egypt; 7Department of Medical Laboratory Sciences, College of Applied Medical Sciences, University of Bisha, Bisha 61922, Saudi Arabia; muslah70@gmail.com

**Keywords:** skin, cell culture, stem cells, differentiation, camel

## Abstract

**Simple Summary:**

This is the first comprehensive study to isolate different cellular types and stem-like cells from the camel skin. We reported the multipotency of the isolated stem cells. Moreover, some unique cells were observed, such as dermal cyst-forming cells. This discovery represents a cheap and easy source for camel stem cells that is essential for development of the elite camel regenerative medicine and provides a good source of camel fibroblast required for camel cloning.

**Abstract:**

Elite camels often suffer from massive injuries. Thus, there is a pivotal need for a cheap and readily available regenerative medicine source. We isolated novel stem-like cells from camel skin and investigated their multipotency and resistance against various stresses. Skin samples were isolated from ears of five camels. Fibroblasts, keratinocytes, and spheroid progenitors were extracted. After separation of different cell lines by trypsinization, all cell lines were exposed to heat shock. Then, fibroblasts and dermal cyst-forming cells were examined under cryopreservation. Dermal cyst-forming cells were evaluated for resistance against osmotic pressure. The results revealed that resistance periods against trypsin were 1.5, 4, and 7 min for fibroblasts, keratinocytes, and spheroid progenitors, respectively. Furthermore, complete recovery of different cell lines after heat shock along with the differentiation of spheroid progenitors into neurons was observed. Fibroblasts and spheroid progenitors retained cell proliferation after cryopreservation. Dermal cyst-forming cells regained their normal structure after collapsing by osmotic pressure. The spheroid progenitors incubated in the adipogenic, osteogenic, and neurogenic media differentiated into adipocyte-, osteoblast-, and neuron-like cells, respectively. To the best of our knowledge, we isolated different unique cellular types and stem-like cells from the camel skin and examined their multipotency for the first time.

## 1. Introduction

The Arabian camel (*Camelus dromedarius*) is the best food supplier in the desert areas due to its adaptability to the scarcity of water and food and the temperature change in the desert [1]. Camel racing is a pivotal traditional sport in the Arabian Gulf states. Many camel injuries result from such racing [2]. Therefore, there is an urgent need for more camel stem cell investigations which play a pivotal role in development of tissue regeneration techniques [3].

The desert animals are exposed to high levels of solar radiation and high temperatures; therefore, their skin structures are specially adapted to prevent damage of the tissue proteins. The camel is clearly adapted for a desert lifestyle and its skin is considered as unique among domestic animals. Its epidermis and dermis layers are similar to that of other hairy mammals [4]. However, it differs in the arrangement of the hair and the morphology of the hair follicles from that of other domestic mammals [5].

Fibroblasts are mesenchymal-derived cell types important in several physiological processes such as synthesis of extracellular matrix (ECM), epithelial differentiation, regulation of inflammation, and wound healing [6]. In addition, fibroblasts derived from the skin are frequently used to produce induced pluripotent stem cells (iPSCs) [7], a powerful tool that allows production of other kinds of desired cells, and is now being widely used for disease modeling in in vitro [8,9,10]. Cell culture technology has become a widely used method in biology, medical research, and biomedical applications. Establishing primary cultures of fibroblasts allows researchers to obtain representative cells that have conserved most of their original characteristics and functions, which is an important foundation for further cell biology and cell engineering. Cryopreservation of animal cells is an excellent technique for long-term preservation of animal genetic resources, which is critical to guarantee accurate genomics and genetic analyses [11,12,13]. Skin cells of Dubai camel (Dubca) have been previously isolated and characterized by Klopries, et al. [14] and Gupta, et al. [15]. Moreover, Sharma, et al. [16] were the first to cryopreserve the *Camelus bactrianus* fibroblasts. Fibroblasts are used as a donor cell for generating cloned camel embryos using somatic cell nuclear transfer [17,18].

Mesenchymal stem cells (MSCs), also named mesenchymal stromal cells or medicinal signaling cells [19,20], are multipotent cells with the potential for differentiating into variety of cell types, have become a promising source for novel cell therapeutic applications [21]. Under appropriate conditions, these cells can be induced to differentiate into neurons, myoblasts, cardiomyocytes, adipocytes, chondrocytes, and osteoblasts [20]. Due to their easy isolation, expansion, and broad differentiation potential, MSCs have been widely studied in regenerative medicine and tissue engineering studies for more than a decade [22]. MSCs can be obtained in relatively large numbers from a variety of tissues and organs, such as bone marrow, deciduous teeth, skeletal muscle, cord blood, and various fetal tissues [23,24]. 

Recently, the camel stem cell (SC) research has attracted some attention [25,26,27]. These studies isolated stem cells from cumulus cells [25], adipose tissue [26], and camel embryos [27]. Currently, the main targets of researchers’ interest are epithelial and fibroblast cells of the skin. The elements of the epithelial cells are found mainly in the epidermis and skin appendages, and components of the fibroblasts are found in the dermis and subcutaneous fat. In accordance with the tissue topography, several skin SC niches could be distinguished, and the principal ones were epithelial (epidermal) SC niche, and dermal and adipose niches [28]. Dermal stem cells reside in the lower dermal sheath/dermal cup areas of the hair follicle [29]. Dermal spheres have been isolated from human skin [30], pig [31], and cow [32]. So far, there has been no report on isolation and characterization of skin stem cells and their differentiation potential into mesenchymal lineage in camel. Potential application of stem cells is currently under intense investigation for treatment of a wide range of animal diseases, particularly that in cell-based orthopedic therapies [33]. Therefore, establishment of a standard protocol for isolation, characterization of skin stem cells, and their differentiation into different tissues exhibits many advantages in cell replacement therapy and tissue engineering in camels. These advantages are even more important in dromedary camels (*Camelus dromedarius*), for which bone fractures are a real problem and more prone to accrue [34].

Therefore, in this study we aimed to isolate and characterize the stem cells, fibroblasts, and keratinocytes from camel skin and to investigate the influence of the conventional in vitro cell culture environment on the proliferation and multi-lineage differentiation potential of the cells after serial passages.

## 2. Materials and Methods 

### 2.1. Tissue Sample Collection

Skin tissue samples were obtained from five healthy dromedary camels from a local abattoir after slaughtering. Skin biopsy specimens were aseptically taken from the ear pinna of healthy camels. Ear skins were washed with normal saline (NaCl, 0.9%) and scrubbed with povidone-iodine or chlorhexidine solution, using an impregnated brush to remove any remaining dirt and then rinsed with phosphate-buffered saline (PBS) and dried with a sterile gauze pad. Ear samples were cut with scissors into 1 × 1 cm^2^ pieces, and then they were transferred to the laboratory for further processing in a sterile 50 mL test tube containing 25 mL of sterile DPBS with ten times the normal concentration of a routine antibiotic dose of antibiotic/antimycotic. The collection tube with tissue samples was then transported to the laboratory at 4 °C within 3 h after collection.

### 2.2. Explant Preparation and Culture

For the isolation of skin fibroblasts, ear tissues were prepared and cultured using a previously described protocol, with slight modifications [11,35,36]. In brief, the ear skin biopsies were dried on cellulose filter paper (WhatmanR; Sigma-Aldrich, St. Louis, MO, USA), soaked once in 70% ethanol for 1 min, allowed to dry, and then washed three times with PBS containing ten times the routine antibiotic/antimycotic dose. Tissue fragments were transferred into a 100 mm tissue culture dish using a sterile scalpel. A 5 mm perimeter was trimmed from the edges of the excised skin; the front and back of excised skin samples were then separated. Subcutaneous tissue (loose connective tissue and lobules of fat) was removed and rinsed with washing PBS. Excised samples were first cut with a scalpel into 1-cm-long strips and then cut into pieces approximately 1 to 2 mm^2^ in size (explants), which were placed in DMEM containing ten times the routine antibiotic/antimycotic dose. These skin explants were minced with iris scissors. Samples were centrifuged at 200× *g* for 10 min to remove the supernatant, digested in collagenase solution (DMEM C 10% v/v collagenase type II C 10 antibiotic/antimycotic) in a 60-mm-diameter culture dish, and incubated at 37 °C and under 5% CO_2_ for 21 h. The next day, the explant samples were further washed twice with PBS. After the initial 1 to 3 days of culture, explant samples were incubated in a 60-mm-diameter culture dish with explant medium containing 20% heat-inactivated FCS together with a routine antibiotic dose to protect against microbial contamination.

### 2.3. Monitoring for Outgrowths of Primary Cell Cultures and Different Cell Types

Once collagenase was removed, the cultures were monitored daily under an inverted microscope at low magnification to observe explant dislodging and the overall radial migration of primary cells around the explants, together with monitoring for any microbial or fungal contamination by checking the size, shape, and movement pattern of the particles in the culture at high magnification. If microbial contamination was detected in any dish or flask, the entire contents were immediately discarded. Tissue fragments were covered with a thin film of explant medium and continued to be cultured until the generation of primary outgrowing cells on day 10 of culture. Tissue fragments were collected for further growth and characterization of different cellular types.

### 2.4. Characterization of the Observed Cells (Morphology, Trypsinization, Heat Shock, and Osmotic Challenge)

After treatment of cells with 0.25% trypsin/EDTA and incubation for 10 min, it was observed that fibroblasts dissociated after 1.5 min of trypsinization, keratinocytes dissociated after 4.5 min of trypsinization, and fluid-filled cysts and clumps dissociated after 7 min of trypsinization. Dissociated cells were washed in PBS and cultured in new culture dishes. Fibroblasts, keratinocytes, and spheroid progenitors were exposed to heat shock (45 °C for 2 h). Cystic cell monolayer was dissociated, and the cells were subcultured and exposed to osmotic stress (500 mOsm/L for 3 min), followed by recovery at 300 mOsm/L for 15 min. 

### 2.5. Cell Differentiation in Dermal Spheroids

Individual dermal spheroids were collected and cultured in medium containing adipogenic- or osteogenic- or neurogenic-differentiation factors and compared with spheroids cultured in plain culture medium without differentiation factors [25,27]. Adipogenic differentiation medium comprised DMEM, 10% FBS, 10% horse serum, 1% penicillin-streptomycin, 100 nM dexamethasone, 0.45 mM isobutyl methyl xanthine, 3 mg/mL insulin, and 1 mM rosiglitazone (Novo Nordisk, Bagsvaerd, Denmark). Osteogenic differentiation medium comprised DMEM, 10% FBS, 1% penicillin-streptomycin, 50 mg/mL L-ascorbic acid (Wako Chemicals, Neuss, Germany), 10mM b-glycerophosphate, 10 nM calcitriol (1a, 25-dihydroxyvitamin D3), and 10 nM dexamethasone. While, neurogenic differentiation medium comprised serum-free DMEM-F12 and 10 mM *all*-trans retinoic acid (R2625, Sigma-Aldrich, St. Louis, MI, USA(.

### 2.6. Establishing Secondary Cultures

Once the cells reached confluence, fibroblast cultures surrounding the pieces of skin were further expanded. The explant medium was poured out of the culture flasks and the cell surfaces washed with PBS three times. In these second and third passages, keratinocytes were removed from the cultures by short trypsinization with 0.1% trypsin/EDTA, as previously described [37]. The volume was adjusted as appropriate for different-sized vessels (for T75 culture flasks, we added 1.0 mL of 0.1% trypsin/EDTA), followed by incubation at 37 °C. After a few minutes (2–3 min), when observed under a microscope, fibroblasts became rounded and began to detach from the plastic surface of the culture flask due to the difference in surface characteristics of keratinocytes and fibroblasts, thus enabling separation of the two cell populations.

To dislocate the remaining loosely attached cells, the flask was tilted to distribute the trypsin evenly over the culture surface. Separated fibroblasts were transferred to a sterile 15 mL centrifuge tube, followed by immediate addition of at least five volumes of fresh growth medium containing 10% FBS and re-cultivating to expand cell numbers in a new T75 culture flask. Cultures were grown at 37 °C in a humidified atmosphere containing 5% CO_2_. During this time, keratinocytes were still attached to the plastic bottom and they preserved their membranous shapes, as seen before trypsinization. To culture keratinocytes, cells were incubated at 37 °C for 10 min more to disperse the cells; then, the skin explants were removed, and fresh growth medium was added to re-cultivate keratinocytes. Moreover, to obtain a larger number of fibroblasts, explants were left in the first flask for a second round of cell growth and incubated as before, while keratinocytes were separated and moved to a new T75 culture flask instead. The culture medium was changed once every 48 h. The 2nd to 4th passages of cultures of these cells were frozen for long-term storage stock. Fibroblasts in the 4th to 7th passages were used for further experiments.

### 2.7. Cryopreservation and Resuscitation of Fibroblasts

Cryogenic preservation of camel skin fibroblasts is generally the same as cryopreservation of most continuous cell lines—by using common dimethyl sulfoxide (DMSO), which permits long-term storage of cells in liquid nitrogen. After routine trypsinization of cells with 0.25% trypsin/EDTA and incubation for 5 min, cell suspensions were diluted with fresh growth medium and the sample thoroughly mixed. Cells were centrifuged at 200× *g* for 10 min. The pellets were re-suspended in freezing medium consisting of 50% DMEM, 10% DMSO, and 40% FBS (v/v). Cell suspensions were aliquoted into cryogenic storage vials (at approximately 1.5–2.0 × 10^6^ cells/vial) and frozen at −80 °C in a deep freezer overnight. The next day the vials were transferred to a liquid nitrogen tank for long-term storage until used for further experiments. To resuscitate fibroblasts or determine post-cryopreservation cell viability, the cells in the frozen vials were quickly thawed at 37 °C and promptly mixed with 6.0 mL of growth medium in a sterile centrifugal tube and then cultured in T75 culture flasks without centrifugation. Viable cells were counted manually using a hemocytometer and expressed as the % of live cells from the total cell count. The next day, attached cells were washed twice with PBS and new growth medium added.

### 2.8. Measuring Cell Viability 

Cells were counted to assess their viability using a traditional cell-counting method, with a bright-line hemocytometer and trypan blue dye [38]. Briefly, cells were first trypsinized with a routine subculture. Cell suspension was mixed in 1:1 ratio with 0.4% trypan blue solution, and incubated for 5 min at room temperature. Then, 20 mL of the cell suspension was applied between the cover slip and the edge of the hemocytometer chamber and examined immediately under a microscope following the method previously described [38].

## 3. Results

### 3.1. Camel Skin Explants and Outgrowths at P0 (Passage Zero)

Camel skin explant displayed early outgrowth of monolayer of keratinocytes on day 10 of explant seeding (Figure 1A). Additional growth of keratinocytes and migration and appearance of cell clumps (which were later characterized as spheroid progenitors) were seen. Furthermore, fibroblasts were either scattered or in colony (Figure 1B–D).

### 3.2. Long-Term Culture of Camel Skin Explants

Explants were incubated in culture medium for 21 days, and then the culture medium was changed after every two days for a further 28 days. On day 21, the explants were distinguishable into fibroblasts and keratinocytes (Figure 2A), and appearance of small and big fluid-filled cysts, referred to as spheroid progenitors, was noticed (Figure 2B). Moreover, cuboidal cells with melanin crystals were also observed (Figure 2C). On day 28, fibroblasts reached the confluency (Figure 2D) and the fluid-filled cysts increased in both number and size (Figure 2E). The formation of an epithelial-like sheet with uniform cuboidal cells was also observed (Figure 2F).

### 3.3. Sensitivity of Different Skin-Derived Cells to Trypsinization

Primary cell culture incubated with trypsin-EDTA for 5 min exhibited varied behavior. Fibroblasts, with round cells, were observed after 1.5 min of trypsinization (Figure 3A). Keratinocytes were distorted after 4.5 min (Figure 3B). The fluid-filled cysts and cell clump layer displayed resistance to trypsin for 5 min, indicating no change in cell morphology after incubation with trypsin (Figure 3C,D); however, they dissociated after 7 min.

### 3.4. Sensitivity of Different Trypsin-Sensitive Skin-Derived Cells to Heat Shock

Dissociated fibroblasts, keratinocytes, and fluid filled cysts and clumps were collected after 1.5, 4.5, and 7 min of trypsinization, respectively, washed in PBS, and cultured in new culture dishes. Fibroblasts, keratinocytes, and spheroid progenitors were exposed to heat shock (45 °C for 2 h). Fibroblast architecture was disturbed (Figure 4A), and the architecture of keratinocytes was partially distorted (Figure 4B). However, spheroid progenitors and the surrounding cells were resistant to heat shock (Figure 4C). After recovery at 38 °C for 4 days, fibroblasts recovered and reached confluency (Figure 4(A3)) and keratinocytes recovered with increased nuclear size (Figure 4(B3)). Furthermore, spheroid progenitors recovered with spontaneous differentiation into neuron-like axons (Figure 4(C3)).

### 3.5. Fibroblast Subculture and Cryopreservation

Fibroblasts with spindle morphology at passage 1 (P1) at 70% confluency (Figure 5A,B) were isolated morphologically and on the basis of sensitivity to trypsin were subcultured to reach 90% confluency and then harvested and cryopreserved in liquid nitrogen. The thawed and resurrected fibroblasts at 90% confluency with clear spindle morphology are shown in Figure 5C,D.

### 3.6. Cystic Cells Subculture, Cryopreservation, and Characterization

Cystic cell monolayer was dissociated and subcultured (Figure 6A). At passage 1 (P1), groups of cells with cysts and individual cells with different fluid-filled compartments were formed (Figure 6B). After cryopreservation of passage 3 (P3), cystic cells with different number of fluid-filled compartments continued to form (Figure 6D). Exposure of the cystic cells to osmotic stress (500 mOsm/L for 3 min) showed collapse (Figure 6(E1)) of the cyst, which was reformed after recovery at 300 mOsm/L for 15 min (Figure 6(E2)).

### 3.7. Dermal Spheroid Characterization 

Cell clumps or dermal spheroids were clearly observed either in primary culture (Figure 7A) and in P1 (Figure 7B) in form of solid clumps or fluid-filled clumps with resemblance to the embryoid bodies as observed in pluripotent stem cell culture. Individual spheroid exhibited uniform nuclear assembly of over than 100 cells, as observed using Hoechst staining (Figure 7(C1)).

### 3.8. Dermal Spheroid Differentiation

Individual dermal spheroids were collected and cultured in plain culture medium, exhibited formation of spindle-shaped cells (Figure 8A). Spheroids were cultured in medium containing 10 µg/mL of all-trans retinoic acid for inducing neurogenic differentiation with axon-like formation (Figure 8B). Spheroids, cultured in medium containing adipogenic-differentiation factors, showed formation of lipid droplets uniformly distributed in polygonal cells (Figure 8C) after 10 days of culture and were positive for oil-red staining after 18 days of culture (Figure 8D). Spheroids cultured in a medium containing osteogenic-differentiating factors showed positively stained mineralization after Alizarin red staining when compared with those cultured in plain medium without the differentiation factors (Figure 8F).

## 4. Discussion

Stem cell research represents a promising therapeutic strategy for injured, traumatic, or fractured genetically superior camels, which are utilized for racing. In this study, we isolated camel dermal stem cells using trypsin digestion and demonstrated that camel dermal sphere cells are a cheap and readily available source of multipotent stem cells that can differentiate into neuron-, osteoblast-, and adipocyte-like cells which may be utilized in cases which require self-regenerative tissues. Dermal spheres have been isolated from other species including rodents [39], pigs [40] and human [41], and showed multipotency and differentiation capabilities into different cells such as neurons, glia, and smooth muscle cells.

Dermal stem cells have high potential in regenerative medicine for nerves, spinal cord, heart, and other applications. The present study suggested that the potential of dermal stem cells is maintained even after their cryopreservation. Therefore, a bio-bank that contains variable camel dermal stem cells is desired [42]. Our experiment revealed complete maintenance of fibroblast functions after cryopreservation, which indicated preservation of good quality fibroblast isolated from elite camels for cloning and reprogramming. A similar successful cryopreservation result has previously been obtained for the fibroblasts of *Camelus bactrianus* [16]. In addition, fibroblasts derived from skin can be reprogrammed to produce iPSCs through transfection with pluripotency factors (Oct4, Klf4, Sox2, and myc) [7]. These cells could be differentiated into other kinds of cells, and they are widely used for disease modeling in in vitro [8,9,10].

Our results revealed that the strongest attachment was exhibited by the dermal spheroids. The trypsin treatment significantly reduced CD44+, CD55+, CD73+, CD105+, CD140a+, CD140b+, and CD201+ cell number within 30 min [43]. The different trypsin resistances of fibroblasts, keratinocytes, and dermal spheroids may be attributed to varied expression of surface attachment molecules on the surface of the different cells.

The detection of dermal cysts is an impressive finding in camel skin culture. The current results could be considered the first study reporting this finding. Bernerd, et al. [44] observed dermal cysts in the rhino mouse. The authors concluded that cysts of the rhino mouse exhibit high similarity with sebaceous glands and outer root sheath cells. These structures can be easily isolated, and could, therefore, serve as a closed sebaceous gland. Their high resistance to cryopreservation and high pressure could be attributed to their sebum content. Further investigation is warranted to detect the mechanism of protection against high pressure and cryopreservation.

Interestingly, our results revealed, for the first time, the differentiation of spheroid progenitors into neurons after heat shock recovery. Similar results were reported by Nørgaard, et al. [45] who observed the differentiation of human mesenchymal stem cells (hMSCs) into osteoblastic cells after heat shock. Chen, et al. [46] also reported the induction of osteoblasts differentiation after periodical heat shock. They attributed the differentiation to mild heat stress-induced elevation of levels of heat shock proteins, which enhances the responsiveness of stem cells to differentiate. Further studies are needed to reveal the relationship between levels of heat shock proteins and the genes responsible for the neuron differentiation. The ability of fibroblast and keratinocytes to tolerate the heat shock may be attributed to the generation of low levels of reactive oxygen species in the camel fibroblast. A similar result was observed in the Tharparkar cattle breed which resides in similar heat conditions as camels [47]. The authors related the heat shock tolerance to the low expression of HSPA1A and HSPA2 in the dermal fibroblasts of Tharparkar cattle. The high resistance of the spheroid progenitor to the heat shock may be attributed to the increase in levels of heat shock proteins similar to cumulus cells [48] which exhibit multipotency [25] like spheroid progenitor.

At the beginning of the 21st century, researchers succeeded in revealing multipotent progenitor cells lines in the dermis. According to several studies, these cells were found among the fraction of fibroblast-like cells of the primary dermis cell suspension which were adherent to the plastic [49]. In other experiments, vice versa, the progenitor cells were identified among the non-adherent cells of the dermis [50]. This led to the hypothesis that the dermis contained two subpopulations of multipotent cells maintaining its homeostasis and capable of mesodermal differentiation, namely, adhesive multipotent mesenchymal stromal cells (MSCs), being the derivatives of embryonic mesenchyma, and non-adherent in primary culture multipotent progenitor cells of skin (Skin-derived precursors, SKPs), originating from the embryonic neural crest [51].

Generally accepted sources of multipotent SCs of the dermis are dermal papilla and hair sac of HF from which they can be derived during in vitro culturing. Adhesive fibroblast-like cells, migrated from single hair follicles of an adult, possess a high proliferative potential. In appropriate induction conditions, such cells start to differentiate into osteogenic, adipogenic, chondrogenic, or myogenic lineages whereas more than 70% of the SC clones exhibit bi- or trilinear differentiation potential [52]. In other words, it was proved that the adhesive stromal cells of the hair follicle of an adult were truly multipotent mesenchymal SCs and did not represent a heterogeneous population of progenitor cells with limited capacity for differentiation. Another type of dermal multipotent cells derived from the HF dermal papilla, the SKPs, possesses the ability to follow not only mesodermal but several neural differentiation directions as well. Unlike MSCs, when cultured in the presence of growth factors, the primary SKPs do not adhere to the plastic but form spheroids [53].

SKPs isolated from HF dermal papillae can produce dermal cells and induce hair morphogenesis like MSCs [54]. They are also capable of differentiating into neurons without selection and expansion, and under induction conditions, they produce an offspring with phenotypes similar to those of adipocytes, smooth muscle cells, and the cells of the peripheral nervous system, such as Schwann cells and catecholaminergic neurons. The bulge area is the major hair follicle source of nestin-expressing pluripotent stem cells which can repair the spinal cord unlike dermal papilla [55].

Porcine skin-derived stem cells have been demonstrated to exhibit the intrinsic ability to differentiate into oocyte-like and primordial germ-like cells in in vitro [56]. The oocyte-like cells can spontaneously develop into parthenogenetic embryo-like structures. However, they failed to be fertilized by sperm in in vitro because of their unstable structure. Until now, this germline potential of skin-derived stem cells has not been confirmed by other independent groups even though porcine SKP cells can potentially integrate into the genital ridge in in vivo when injected into peri-morula embryos [57]. A more stringent assay, such as production of a chimeric animal, should be performed to evaluate the germline potency of porcine skin-derived stem cells.

Multilineage differentiation potentiality has been considered as an important trait of MSCs. In in vitro, under the action of some inducing factors, Dermal MSCs can differentiate into adipogenic, osteogenic, and other lineages [58]. However, the mechanisms and factors of differentiation need further investigation. In our study, we differentiated camel DMSCs into osteoblasts, adipocytes, and neurons. In addition, Zhao, et al. [59] demonstrated that pig dermis-derived MSCs could be induced to differentiate into osteoblasts and adipocytes.

MSC populations obtained from different tissues exhibit significant differences in their proliferation, differentiation, and molecular properties, which should be taken into consideration when planning their use in clinical protocols [60]. Further investigations for elucidation of molecular characteristics of skin stem cells are required with respect to adipose tissue stem cells [26] and cumulus cells [25].

## 5. Conclusions

For the first time, we isolated different differentiated cells (fibroblasts and keratinocytes), cyst-forming cells, in addition to multipotent dermal stem cells from the camel dermis. The stem cells were capable of forming spheres, which formed neurons, osteoblasts, and adipocytes. These data strongly indicated that stem cells in the dermis constitute a reservoir for camel skin repair elements. Thus, our work provides an opportunity to explore the involvement of dermal stem cells in physiological and pathological events, such as tissue repair. Further studies are essentially needed to isolate and propagate different skin-derived cells from camel based on cell surface markers, however, the commercially available antibodies specific for camel cellular cluster of differentiation (CD) are currently lacking.

## Figures and Tables

**Figure 1 animals-09-00378-f001:**
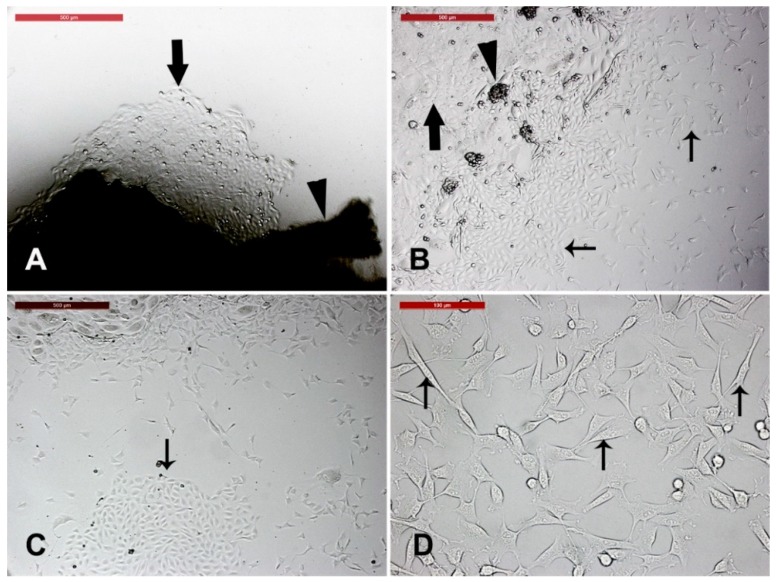
Camel skin explants and outgrowths at P0 (passage zero). (**A**) Camel skin explant (arrowhead) with early outgrowth of monolayer of keratinocytes (black arrow) on day 10 of explant seeding. (**B**,**C**) Additional growth of keratinocytes (black arrow) and migration and appearance of cell clumps (arrowhead; later characterized as spheroid progenitors) fibroblast either scattered or in colony (thin arrows). Scale bars = 200 µm. (**D**) A magnified image of the fibroblast (thin arrows). Scale bar = 100 µm. All cultures were performed in 35 mm tissue culture dishes and maintained in the same conditions in humidified atmosphere at 38 °C under 5% CO_2_.

**Figure 2 animals-09-00378-f002:**
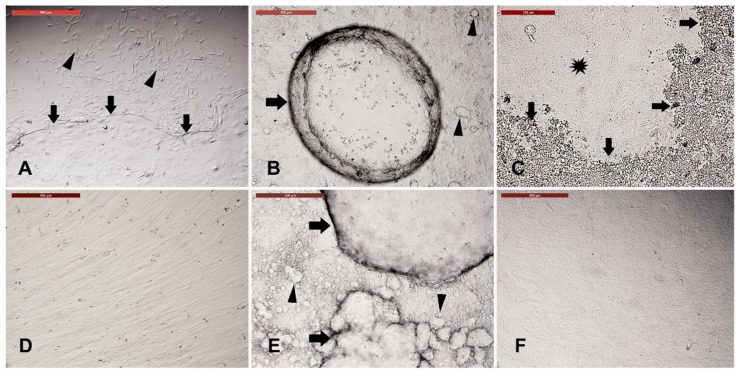
Long-term culture of camel skin explants. Explants were cultured for 21 (**A**–**C**), and further, for 28 (**D**–**F**) days while changing the culture medium after every two days. (**A**) Distinguishing fibroblasts (black arrow) from keratinocytes (white arrow). (**B**) Appearance of small (arrowheads) and big (arrow) fluid-filled cysts. (**C**) Appearance of cuboidal cells (asterisk) with melanin crystals (arrows) (Scale bar = 100 µm). (**D**) Fibroblasts reached confluency on day 28. (**E**) The fluid-filled cysts increased in both number (arrowheads) and size (arrows). (**F**) Formation of epithelial-like sheet with uniform cuboidal cells. Scale bars for all, except (**C**) = 500 µm.

**Figure 3 animals-09-00378-f003:**
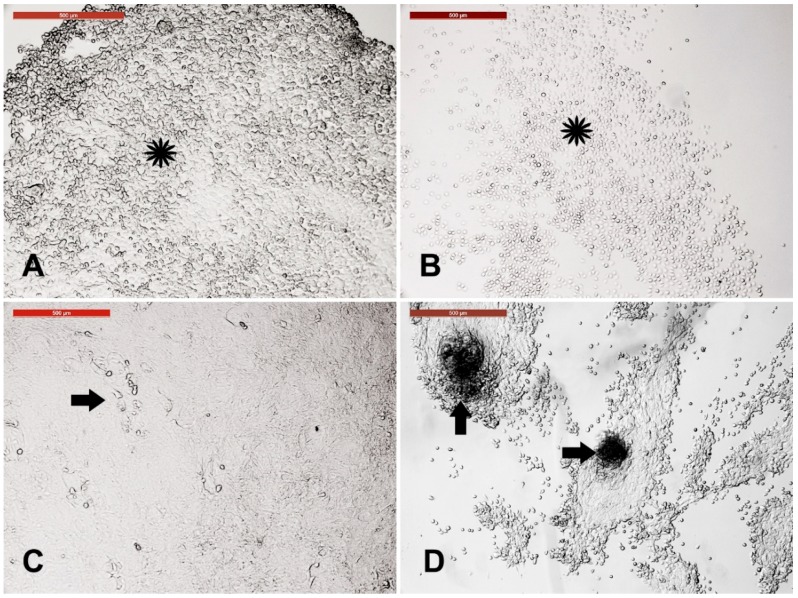
Sensitivity of different skin-derived cells to trypsinization. Incubation of primary cell culture with trypsin-EDTA for 5 min showed varied behavior. (**A**) Keratinocytes, the cell sheet, was distorted after 4.5 min (asterisk). (**B**) Fibroblasts, with round cells, were observed after 1.5 min of trypsinization (asterisk). (**C**) The fluid filled cyst layer showed resistance to trypsin for 5 min (arrow) indicating no change in cell morphology after trypsinization. (**D**) Cell clumps (spheroids) showed resistance to trypsin and the surrounding cells (arrows). Scale bar = 500 µm.

**Figure 4 animals-09-00378-f004:**
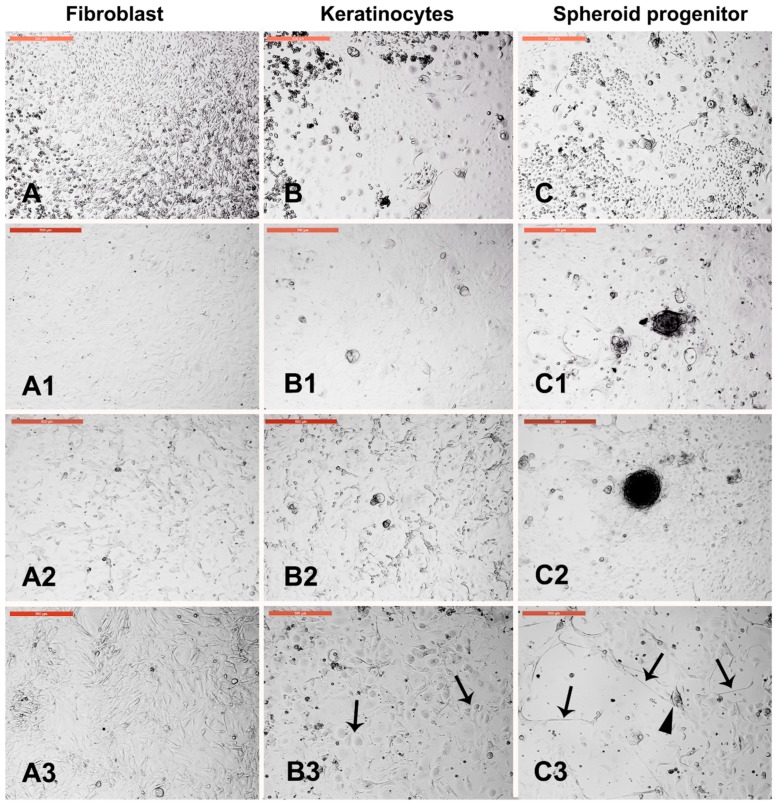
Sensitivity of different trypsin-sensitive skin-derived cells to heat shock. Dissociated fibroblasts (**A**), keratinocytes (**B**), and fluid filled cysts and clumps (**C**) were collected after 1.5, 4.5, and 7 min of trypsinization, respectively, washed in PBS, and cultured in new culture dishes. (**A1**) Fibroblast monolayer. (**B1**) Keratinocyte monolayer. (**C1**) Fluid-filled cysts and cell clumps (spheroid progenitors). Cells of (**A1**,**B1**,**C1**) were exposed to heat shock (45 °C for 2 h). (**A2**) Fibroblast architecture was disturbed. (**B2**) Architecture of keratinocytes was partially distorted. (**C2**) Spheroid progenitors and the surrounding cells were resistant to heat shock. (**A3**,**B3**,**C3**) show the corresponding cells after recovery at 38 °C for 4 days. (**A3**) Fibroblasts recovered and reached confluency. (**B3**) Keratinocytes recovered with increased nuclear size (arrows). (**C3**) Spheroid progenitors recovered with spontaneous differentiation into neuron-like axons (arrows) and cell bodies (arrowheads). Scale bar = 500 µm.

**Figure 5 animals-09-00378-f005:**
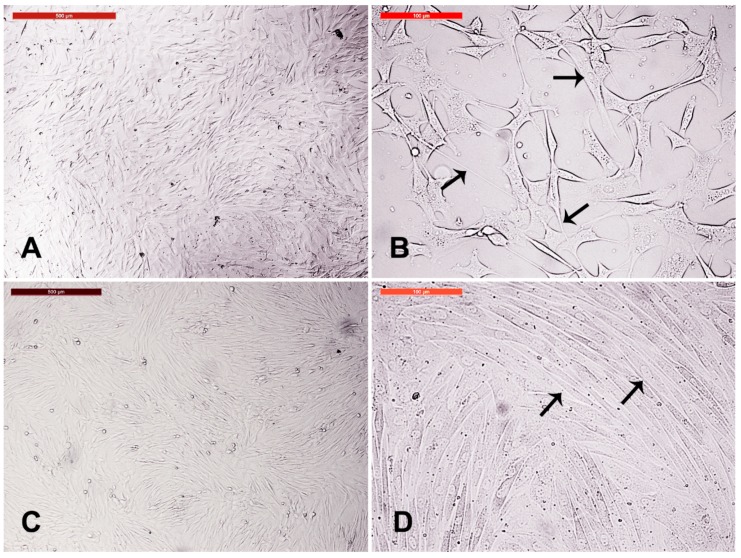
Fibroblast subculture and cryopreservation. Fibroblast were isolated morphologically and based in sensitivity to trypsin, were subcultured, and cryopreserved. (**A**) Fibroblasts at passage 1 (P1) at 70% confluency, and (**B**) magnified image to show spindle morphology (arrows). Cells were cultured to reach 90% confluency, harvested, and cryopreserved in liquid nitrogen. (**C**,**D**) Thawed and resurrected fibroblast at 90% confluency with clear spindle morphology (arrows). Scale bar of (**A**,**C**) = 500 µm. Scale bar of (**B**,**D**) = 100 µm.

**Figure 6 animals-09-00378-f006:**
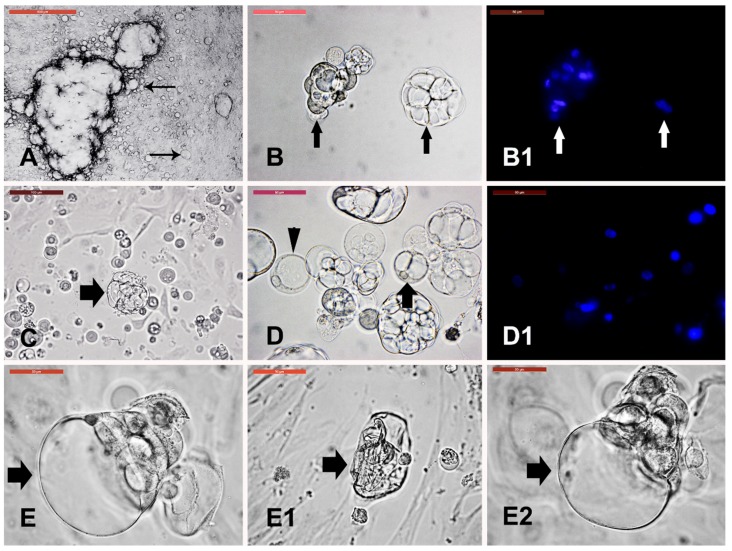
Cystic cells subculture, cryopreservation, and characterization. (**A**) Cystic cell (arrows) monolayer was dissociated and cells were subcultured. (**B**) Groups of cells with cysts and individual cells with different fluid-filled compartments (arrows) at passage 1 (P1). (**B1**) Hoechst staining of nuclei of groups in (**B**). (**C**) Consistent formation of cystic cells (arrow) after cryopreservation of passage 3 (P3) with different numbers of fluid-filled compartments. (**D**) Single (arrowhead); double (arrow); and more than two compartments as distributed in the field. (**D1**) Hoechst staining of nuclei of cells shown in (**D**). (**E**) Exposure of the cystic cells (arrow) to osmotic stress (500 mOsm/L for 3 min) showed cyst collapse ((**E1**), arrow), which was reformed after recovery ((**E2**), arrow) at 300 mOsm/L for 15 min. Scale bar = 50 µm except for (**A**,**C**) are 500 µm.

**Figure 7 animals-09-00378-f007:**
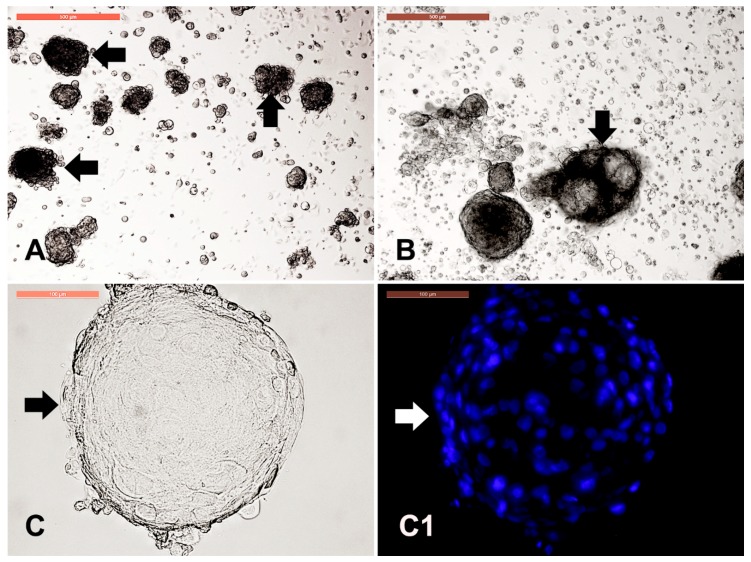
Dermal spheroid characterization. Cell clumps or dermal spheroids were clearly observed either in primary culture (P0) and in P1 in form of solid clumps ((**A**), arrows) or fluid-filled clumps ((**B**), arrow) with resemblance to the embryoid bodies as observed in pluripotent stem cell culture. Scale bar = 500 µm. Individual spheroid ((**C**), arrow) were stained with Hoechst to reveal the uniform nuclear assembly of over than 100 cells to form the spheroids (**C1**). Scale bar = 100 µm.

**Figure 8 animals-09-00378-f008:**
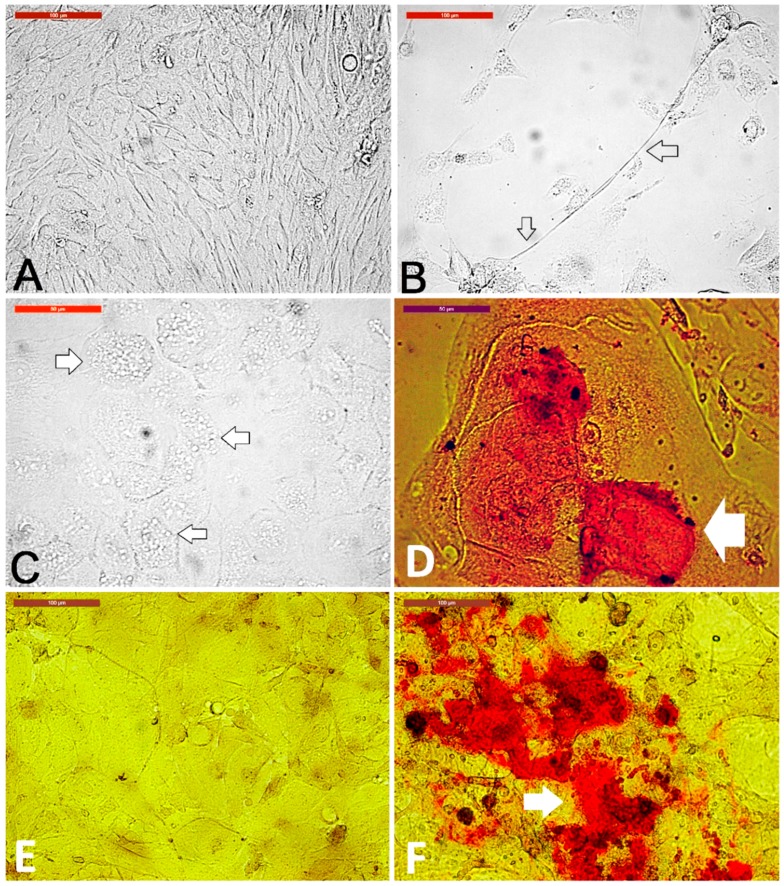
Dermal spheroid differentiation. Individual dermal spheroids were collected and cultured in plain medium (**A**) showed formation of spindle-shaped cells. Scale bar = 100 µm. Spheroids were cultured in a medium containing 10 µg/mL of all-trans retinoic acid for neurogenic differentiation (**B**) with axon-like formation (arrows). Scale bar = 100 µm. Spheroids cultured in medium containing adipogenic-differentiation factors showed formation of lipid droplets uniformly distributed in polygonal cells after 10 days of culture (**C**) and were positive for oil-red stain after 18 days of culture ((**D**), red stain, arrow). Spheroids cultured in medium without differentiation factors (**E**) compared to spheroids cultured in a medium containing osteogenic-differentiation factors showed positively stained mineralization after Alizarin red staining ((**F**), red stain, arrow). Scale bar = 100 µm.

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
