# Peer review of "Isolation and Culture of Skin-Derived Differentiated and Stem-Like Cells Obtained from the Arabian Camel (Camelus dromedarius)"

_animals, 2019, doi:10.3390/ani9060378_

Round 1
Reviewer 1 Report
In the present study, the authors aimed to isolate and characterize the stem cells, fibroblasts, and keratinocytes from camel skin, and to investigate the influence of the conventional in vitro cell culture environment on the proliferation and multi-lineage differentiation potential of the cells after serial passages.
This article is interesting and it highlights an useful topic. All the experiments were good executed. The english is adequate.
I suggest, in the discussion section, to stress more the concept of induced pluripotent stem cells (iPSCs), and the falls within this research.
Author Response
Comment:
In the present study, the authors aimed to isolate and characterize the stem cells, fibroblasts, and keratinocytes from camel skin, and to investigate the influence of the conventional in vitro cellculture environment on the proliferation and multi-lineage differentiation potential of the cells afterserial passages.
This article is interesting and it highlights an useful topic. All the experiments were good executed. The english is adequate.
I suggest, in the discussion section, to stress more the concept of induced pluripotent stem cells (iPSCs), and the falls within this research.
Response:
We appreciate the time, efforts, and suggestions by the reviewer. All the suggestions have been addressed, and corrections made accordingly. We added some discussion related to this point in the revised manuscript.
Reviewer 2 Report
Saadeldin et al. isolate and characterize dermal stem-like cells from camels, a cell population of potential interest in regenerative approaches for racing animals. The article merits publication in Animals but I have a number of suggestions that in my opinion would improve the manuscript.
1- Classification of cell types is solely based on morphological features, which can be misleading, and separation of cell fractions is achieved by differential times of trypsin exposure. This technique will produce mixed populations of cells. These drawbacks should be explicitly reflected upon in the discussion.
2- Throughout the article, dermal cysts and spheroid progenitors are discussed together and I found this somewhat confusing. Dermal cysts do express cytokeratins and may represent follicular infundibulum/sebaceous gland primordia in culture. On the other hand, spheroid progenitors are of mesenchymal origin and should not contain any sebum. If the authors cannot provide more experimental evidence on how these structures are formed and composed, I would suggest separating both phenomena in the description and conclusions.
3- Claims of differentiation to neuronal, adipocytic and osteogenic lineages are based on very thin data. In the absence of specific functional markers, transcriptomics and quantification, I would suggest to tone down the wording and state differentiation capacities in a very speculative manner.
4- Exact composition of cell culture media must be fully disclosed (avoid terms such as "blank medium" and references to other papers).
Other relatively minor (but important) suggestions:
- the word differon is very unusual, please explain or remove
- Differentiation potential of MSCs as stated in page 2, line 78 (ref. 20) is widely disputed. See for instance doi:10.1038/nm.3028
- Page 2, line 90. Reference 27 refers to melanocyte stem cells that reside in the bulge. Dermal stem cells reside in the lower dermal sheath/dermal cup areas of the hair follicle.
- When mentioning species where dermal spheres have been isolated, many references are missing, specially those from rodents (mouse and rats) but also from human (doi:10.1634/stemcells.2004-0134), porcine (doi: 10.4236/scd.2013.31011), etc. Please revisit the literature and quote other studies.
Author Response
Comments and Suggestions for Authors:
Saadeldin et al. isolate and characterize dermal stem-like cells from camels, a cell population of potential interest in regenerative approaches for racing animals. The article merits publication in Animals but I have a number of suggestions that in my opinion would improve the manuscript.
Response: We appreciate the time, efforts, and overall constructive criticism raised by the reviewer that greatly contributed to the improvement of manuscript quality. All the suggestions have been addressed, and corrections made accordingly.
Q1- Classification of cell types is solely based on morphological features, which can be misleading, and separation of cell fractions is achieved by differential times of trypsin exposure. This technique will produce mixed populations of cells. These drawbacks should be explicitly reflected upon in the discussion.
R1- We agree with the reviewer for this point. However, the work with camel cells still in the premature stage. Very few commercial antibodies are available to differentially select the cells based on cell markers specific for camel cells, labeling, and flowcytometery.
<Ref: Hussen et al. Open Vet J. 2017; 7: 150–153. doi: 10.4314/ovj.v7i2.12>
In our work we tried to use a simple method for primary separation of cells based on cell morphology, trypsin exposure, and heat shock challenge. Additionally, subculture, cryopreservation and maintenance of the cell morphology were achieved after these processes. We are looking for further studies using advanced analysis tools to select the different camel cell types. We clarified this part in the discussion and conclusion.
Q2- Throughout the article, dermal cysts and spheroid progenitors are discussed together and I found this somewhat confusing. Dermal cysts do express cytokeratins and may represent follicular infundibulum/sebaceous gland primordia in culture. On the other hand, spheroid progenitors are of mesenchymal origin and should not contain any sebum. If the authors cannot provide more experimental evidence on how these structures are formed and composed, I would suggest separating both phenomena in the description and conclusions.
R2- We thank the reviewer for this suggestion. We modified the text and separated the discussion and conclusion accordingly.
Q3- Claims of differentiation to neuronal, adipocytic and osteogenic lineages are based on very thin data. In the absence of specific functional markers, transcriptomics and quantification, I would suggest to tone down the wording and state differentiation capacities in a very speculative manner.
R3-We thank the reviewer for the suggestions. We modified the description of the resultant differentiated cells accordingly.
Q4- Exact composition of cell culture media must be fully disclosed (avoid terms such as "blank medium" and references to other papers).
R4- We added the details of culture media used for differentiation accordingly.
Other relatively minor (but important) suggestions:
Q5- the word differon is very unusual, please explain or remove
R5- We replaced it by the appropriate words accordingly.
Q6- Differentiation potential of MSCs as stated in page 2, line 78 (ref. 20) is widely disputed. See for instance doi:10.1038/nm.3028
R6- We agree with the reviewer regarding the terminology of the MSCs. We provided this citation and another citation to draw the attention of “Animal” audience about this controversy.
Q7- Page 2, line 90. Reference 27 refers to melanocyte stem cells that reside in the bulge. Dermal stem cells reside in the lower dermal sheath/dermal cup areas of the hair follicle.
R7- We thank the reviewer for this notion. We corrected it accordingly.
Q8- When mentioning species where dermal spheres have been isolated, many references are missing, specially those from rodents (mouse and rats) but also from human (doi:10.1634/stemcells.2004-0134), porcine (doi: 10.4236/scd.2013.31011), etc. Please revisit the literature and quote other studies.
R8- We thank the reviewer for the suggestion. We already mentioned some references in Lines 91-92 and we added extra explanation regarding this point as recommended in Lines 340-341.
Round 2
Reviewer 2 Report
-